# Early Administration of Rifampicin Does Not Induce Increased Resistance in Septic Two-Stage Revision Knee and Hip Arthroplasty

**DOI:** 10.3390/antibiotics14060610

**Published:** 2025-06-16

**Authors:** Leonard Grünwald, Benedikt Paul Blersch, Bernd Fink

**Affiliations:** 1Department of Joint Replacement, General and Rheumatic Orthopaedics, Orthopaedic Clinic Markgröningen gGmbH, Kurt-Lindemann-Weg 10, 71706 Markgröningen, Germany; leonard.gruenwald@rkh-gesundheit.de (L.G.); benedikt.blersch@rkh-gesundheit.de (B.P.B.); 2Department of Trauma and Reconstructive Surgery, BG Klinik, University of Tübingen, Schnarrenbergstraße 95, 72076 Tübingen, Germany; 3Orthopaedic Department, University Hospital Hamburg-Eppendorf, Martinistrasse 52, 20246 Hamburg, Germany

**Keywords:** arthroplasty, rifampicin, periprosthetic joint infection, two-stage revision

## Abstract

**Background/Objectives**: Periprosthetic joint infection (PJI) is a severe complication that follows arthroplasty and occurs in approximately 2% of all cases. One of several cornerstones of therapy is an optimized antibiotic regimen. Early administration of rifampicin—together with a combination of an antibiotic to which the specific microorganism is susceptible—accompanying a two-stage revision surgery, remained controversial due to the potential risk of emerging resistance. However, the exact time to start rifampicin treatment often remains unclear and might be crucial in the treatment regimen. **Methods**: In a retrospective study design, a total of 212 patients receiving a two-stage revision surgery after a diagnosis of PJI (60.8% THA, 39.2% TKA) received an individual rifampicin combination therapy after initial debridement and removal of all foreign material, starting rifampicin on the second day postoperatively. **Results**: At the time of spacer explantation, two patients had developed rifampicin resistance (0.9%). At follow-up (M = 55.4 ± 21.8 months) after reimplantation, three patients had developed rifampicin resistance (1.4%). Concerning the development of reinfection, in general, in the study group and the necessity for further treatment, a total of 25 patients showed signs of reinfection (11.8%). **Conclusions**: Only 0.9% after the first stage and 1.4% at follow-up after the second stage of all 212 patients with accompanying long-term rifampicin combination therapy developed a rifampicin resistance. Therefore, rifampicin administration could be started on the second postoperative day when sufficient concentrations of the accompanying antibiotics can be expected.

## 1. Introduction

Periprosthetic joint infection (PJI) is a severe complication that follows arthroplasty and occurs in approximately 2% of all patients undergoing the procedure, and the incidence even seems to be rising [1]. More than 25% of all necessary revision surgeries can be attributed to PJI [2]. The genesis of PJI is multifactorial, including such factors as increased body mass index (BMI), increased age, female gender, several comorbid disorders (i.e., diabetes mellitus) and previous joint surgeries, amongst others [3]. A distinction is made between acute infections and chronic infections. Especially for chronic infections, revision surgery is mandatory and includes the removal of all foreign material, including the implant itself [4]. One-stage or two-stage exchange procedures are the methods of choice, with the success rate of both procedures being almost similar, as ratings in the arthroplasty registers prove [5,6].

Besides the removal of all foreign material and radical debridement, the second cornerstone of treatment is antibiotic therapy. The antibiotic regimen is usually a combination of locally administered antibiotics and long-term targeted systemic antibiotic therapy, depending on laboratory analysis [7]. For systemic antibiotic therapy, a treatment duration of 4–6 weeks up to 3 months following removal of the implant is advisable [1]. There are different antibiotics available, according to the susceptibility of the microorganism causing the PJI. An aggravating factor in the treatment of PJI is that the dosage of antibiotics needs to be 10,000 times higher to have a curative impact on bacteria in an established biofilm than it has to be in the treatment of their free-floating counterparts [8]. The formation of a biofilm on an implant usually takes a few days [9,10].

One of the most important antibiotics in the therapy of PJI is rifampicin because it is one of the few antibiotics that can penetrate the young biofilm [11,12]. Therefore, rifampicin is recommended in the treatment of PJI, especially in staphylococcal PJI [11,12]. Zimmerli and colleagues demonstrated the efficacy of rifampicin in biofilm-associated staphylococcal infections [13]. It should always be accompanied by a companion medication to which the microorganism isolate is susceptible, due to the low barrier of rifampicin in the development of resistance [1,11,14].

However, the optimum time to start the application remains controversial, due to the risk of resistance. The risk of resistance against rifampicin is highest when the concentration of the accompanying antibiotics is low and the burden of bacteria is high. Therefore, the start of therapy is recommended only after debridement and even removal of drains [1]. In a recent study, Beldman and colleagues reported that the treatment outcome was significantly better when rifampicin was only started 5 days postoperatively. However, the authors stated that patients who received rifampicin immediately after surgery (or within the first 5 days) suffered from more severe infections [14]. Unfortunately, the study did not collect data on suspected rifampicin resistance in the follow-up. On the other hand, the delay in starting with rifampicin may allow the remaining bacteria to build a new biofilm on the new implant (in one-stage revision) or spacer (in two-stage revision). Unfortunately, the mature biofilm is less sensitive to antibiotic treatment [10].

To address these uncertainties in the administration of rifampicin early after resection surgery, the following retrospective study addressed the question of the development of rifampicin resistances and therapy success after two-stage exchange procedures with accompanying long-term rifampicin combination therapy starting on the second day postoperatively in patients with chronic PJI.

## 2. Results

A total of 212 patients with confirmed PJI received an accompanying rifampicin treatment, starting on the second postoperative day. A total of 7 patients (3.4%) received rifampicin only intravenously, whereas a total of 34 patients (16.6%) received it only orally. The majority of patients (171, 83.4%) received rifampicin as a combination of both.

Concerning the general reinfection rate—independent of the triggering factor—at the time of spacer explantation (second stage), seven patients (three female, four male) showed persistent infection parameters (3.3%) in blood values and microbiological analysis. A total of 205 patients did not show infection parameters anymore (96.7%). At follow-up after reimplantation (M = 55.4 ± 21.8 months) 25 patients (11.8%) showed reinfection and had to be treated with two-stage re-revision again, whereas 187 showed no signs of reinfection (88.2%).

Specifically analyzing potential rifampicin resistances at the time of spacer explantation, two of those seven patients (one male, one female) with proof of microbial infection—both underwent TKA—developed rifampicin resistance (0.9%). Both patients received a combination therapy of rifampicin orally and intravenously and did not have any other revision surgery before this two-stage exchange procedure. One patient initially showed a monobacterial infection with *Streptococcus agalactiae* that shifted to *Staphylococcus epidermidis* (rifampicin resistant), and the other case initially showed a monobacterial infection with *Staphylococcus aureus* that shifted to *Staphylococcus capitis* (rifampicin resistant)

The mean age of the two patients with rifampicin resistance was 50.76 ± 14.86, whereas those without resistance were on average twenty years older (69.37 ± 11.23). This difference reached statistical significance (F = 5.429; *p* = 0.021). Concerning BMI, patients with developed resistance showed significantly higher values with 49.38 ± 14.30 versus 30.25 ± 6.64 (F = 16.148; *p* < 0.001).

At follow-up after reimplantation (M= 55.4 ± 21.8 months), three patients were detected with rifampicin-resistant microorganisms (one female, two males). Therefore, for follow-up in the study sample, a total of 1.4% developed rifampicin resistance.

There was no overlap between patients with rifampicin-resistant organisms at spacer explantation and at follow-up. One of the two patients with rifampicin resistance at the time of the first reimplantation showed reinfection with a pathogen shift to a rifampicin-sensible *Staphyloccocus aureus.* The schematic study course is shown in Figure 1.

For the follow-up sample, no significant effects of age or BMI were found. BMI was slightly higher for patients with developed resistance (33.2 ± 10.4 vs. 30.4 ± 6.9), but the difference did not reach statistical significance. Concerning preceding surgeries, two patients with rifampicin resistance had already had septic revision surgery before the initial two-stage revision surgery described in this study. However, there was no microbiological or histological information available.

## 3. Discussion

This study showed that only a small percentage of patients with accompanying long-term rifampicin combination therapy developed a rifampicin resistance: only two in the described sample, which corresponds to a percentage of 0.9% at the time of spacer explantation and three in long-term follow up after reimplantation (1.4%). Particularly important is that rifampicin administration started as early as possible, which means on the second postoperative day when it could be expected that the identified companion medication reached therapeutic effectiveness. These results are in line with a randomized controlled trial study with patients suffering from PJI, who were treated with DAIR: a total of 23 patients received a rifampicin combination therapy with rifampicin started on day 1 after surgery, whilst the companion antibiotic was started perioperatively. In this rifampicin-treatment group, the two patients suffering from treatment failure did not develop rifampicin resistance [15]. Related to these results, a large multicenter retrospective study with a study population of 374 patients found evidence for the benefits of rifampicin use in patients with chronic PJI caused by *S. aureus* who were treated with two-stage exchange arthroplasty. Moreover, the authors describe that only one patient in the rifampicin-treated group (*n* = 187) had a reinfection relapse, due to a developed rifampicin-resistant strain, which equals a percentage of 0.53%. However, due to the study design, it cannot be retraced precisely how the rifampicin regimen after surgery had been conducted and if samples of all patients had been examined for microbiological failures or only those with clinical complaints [16].

In a recently published meta-analysis, the authors concluded that adding rifampicin to monotherapy for patients with staphylococci PJI was associated with a higher likelihood of therapeutic success. However, the authors could not define the best timing to start rifampicin [12]. Most included studies in the meta-analysis started rifampicin only after the bacterial load had been reduced by a companion drug (delay of up to two weeks). However, in original animal studies, rifampicin was able to prevent upcoming infections completely when given as early as possible, which might imply starting medication as soon as possible [17]. In PJI, the delayed start of rifampicin treatment might be due to reports of treatment regimens of endocarditis, where rifampicin resistance had a rate of 21.4% when medication was started before the clearance of bacteraemia [17]. However, in two-stage exchange procedures of PJI, the radical debridement significantly reduces bacterial load before antibiotics are started. Therefore, the basic conditions in treatment have tremendous differences between endocarditis and PJI—and drawing conclusions from endocarditis treatment to PJI treatment might be ineligible.

In general, studies with a comparative design concerning the specific start of rifampicin administration are rarely found in the literature. In 2021, Darwich and colleagues compared retrospectively two subgroups with chronic and acute PJI: one subgroup with 25 patients received rifampicin only several days delayed (after antibiograms had been obtained), and the other group with 37 patients received rifampicin immediately after surgery (same day) [18]. The second group with immediate rifampicin administration had developed a 7% greater rifampicin resistance (19%) than the other group with 12% after delayed administration. However, the difference to our reported study is that Darwich and colleagues reported on cases with an initial start of rifampicin, whereas in our sample, the start of rifampicin administration was only on day 2 postoperatively, to be sure that the concentration of the accompanying antibiotic is sufficiently high. Therefore, we avoided monotherapy with rifampicin and the related higher risk of developing rifampin resistance [1,11,14]. Therefore, the results of our study with 1.4% long-term rifampicin resistance are even lower than the results of Darwichs second group, with delayed rifampicin administration (12%).

In our analysis, the two patients with rifampicin resistance, initially developed at spacer explantation, were significantly younger and had significantly higher BMI scores than those without resistances. For rifampicin resistances in long-term follow-up after reimplantation, a significant effect of body mass could not be observed anymore, although a difference in average BMI scores was visible. Unfortunately, the sample is too small to draw substantial conclusions. Effects of weight have been documented several times concerning PJI recovery [19], but, to our knowledge, not concerning the development of rifampicin resistances during PJI treatment. But, as the development of rifampicin-resistant strains is one specific possibility of treatment failure, some determining factors could be equal.

Interpretation of significantly younger age patients with rifampicin resistance with spacer explantation seems to be more complex, particularly as the effect is not visible anymore in the follow-up. In their observational study of 664 patients, Beldman et al. reported that, at ages older than 80, there was a significant risk factor for treatment failure; however, for the rifampicin cohort, this effect was not visible anymore. This suggests that, for patients receiving rifampicin, old age did not seem to be a risk factor [14]. For patients who underwent two-stage revision, Krizsan et al. reported a generally older age as a risk factor for clinical failure. But beyond that, they detected a significant interaction between age and rifampicin resistance, indicating that younger patients with rifampicin resistance suffered from poorer results [20]. One hypothesis could be that metabolization of rifampicin might be more effective in younger patients and higher dosages would be necessary. Nguyen et al. reported in their study with 154 analyzed cases no influence of different dosing (≤600 mg to ≥1200 mg) with levofloxacin as a co-antibiotic. However, daily dosages corresponded to body weight but not according to age [21].

The overall success rate of the described treatment process with 88.2% of all patients is in accordance with other studies with accompanying rifampicin administration with a slightly delayed start, as well as general success rates of two-stage revision surgeries [5,11]. Due to the retrospective design of this study, there is no comparison sample available. Therefore, the question of potential shifts in results for an even more delayed start of rifampicin administration cannot be answered. However, Darwich and colleagues did not report significantly better results with more delay in rifampicin administration. Nevertheless, upcoming prospective studies should answer these questions with specific subgroups.

Therefore, one limitation of this study is its retrospective design and the lack of a control group with a delayed beginning of the rifampicin therapy. However, the goal of the study was to analyze if early administration of rifampicin leads to a high rate of rifampicin resistance. A further limitation is that the study only analyzed the frequency of reinfection and rifampicin resistance and no other antibiotics and possible influencing factors (for example, comorbidities, previous revisions). Moreover, only the dosage of 600 mg of rifampicin per day was used and analyzed. Therefore, it is not known if a higher dosage of rifampicin would lead to different rates of reinfection and resistance. Although the sample size is sufficiently high and the main research question concerning the possible development of rifampicin resistances could be answered satisfactorily, the answer to other important questions is difficult to define, with a sample size of only two rifampicin resistances. Therefore, a further study with a prospective design, a control group and a larger sample with the possibility of subgroup formations is desirable.

## 4. Materials and Methods

This is a single-center retrospective cohort study of a tertiary center for treating periprosthetic joint infection (Orthopaedic Clinic Markgröningen): patient data that was captured prospectively in the database of the information system of the Orthopaedic Clinic Markgröningen was retrospectively analyzed. All included patients received rifampicin combination therapy during two-stage revision surgery following confirmed PJI of their THA or TKA between 2018 and 2021. Periprosthetic joint infection was diagnosed preoperatively in all cases according to the criteria of the Musculoskeletal Infection Society (MSIS) and the International Consensus on Musculoskeletal Infection (ICM) 2018 [22,23]. Exclusion criteria were a follow-up of fewer than 24 months and cases with histopathological examination without classification of the periprosthetic membrane, according to Krenn and Morawietz [24,25,26], as well as patients with initially rifampicin-resistant pathogens. The medical records of all patients were reviewed for sex, age, body mass index (BMI), time from primary implantation to revision, American Society of Anesthesiologists (ASAs) score, Charlson Comorbidity Index (CCI) score, comorbidities, prior revision surgeries (septic, aseptic), type of infection (monobacterial or polybacterial), type of pathogen (easy-to-treat, difficult-to-treat (according to published reports) or methicillin-resistant staphylococci), histopathological examination (neutrophil granulocytes per high-power field, classification of the periprosthetic membrane according to Krenn and Morawietz), microbiological examination, laboratory parameters and follow-up (later reinfection or no reinfection).

Statistical analyses were performed as described in Section 4.5.

### 4.1. Patient Characteristics

All 212 patients with PJI (60.8% THA, 39.2% TKA) received a two-stage revision surgery, with an immediate accompanying rifampicin treatment at the second day postoperative. Patients with antibiotic therapy without rifampicin were excluded from the study. The mean age was 69.2 ± 11.4 (27.8–96.8) years, with 124 males (58.5%) and 88 females (41.5%). The body mass index (BMI) of the patient cohort averaged 30.4 ± 6.9 (17.7–60.6) kg/m^2^. Concerning the American Society of Anesthesiologists (ASAs) risk classification, 5 patients (2.4%) were classified as ASA I, 94 patients (44.3%) as ASA II, 107 patients (50.5%) as ASA III, and 6 patients (2.8%) as ASA IV [27,28]. Demographic data are shown in Table 1.

Regarding the Charlson Comorbidity Index (CCI), 12 patients (5.7%) were categorized as CCI 0, 12 patients (5.7%) as CCI 1, 38 patients (17.9%) as CCI 2, 41 patients (19.3%) as CCI 3, 43 patients (20.3%) as CCI 4, 31 patients (14.6%) as CCI 5, 22 patients (10.4%) as CCI 6, 8 patients (3.8%) as CCI 7, 2 patients (0.9%) as CCI 8, 1 patient (0.5%) as CCI 9, 1 patient (0.5%) as CCI 10 and 1 patient (0.5%) as CCI 11. In terms of secondary diseases potentially relevant to the development of PJI, 10 patients (4.7%) had a rheumatic disease and 44 patients (20.8%) had diabetes mellitus [28,29].

The average time between the primary implantation and the subsequent two-stage septic revision was 93.3 ± 80.78 (2–500) months. A septic revision was carried out in 59 patients (27.8%), and 12 patients (5.7%) had aseptic revision prior to the analyzed two-stage septic revision.

### 4.2. Characteristics of PJI

The periprosthetic joint infections were monobacterial in 181 cases (85.4%) and polybacterial in 31 cases (14.6%). Easy-to-treat (ETT) pathogens were found in 172 infections (81.1%), difficult-to-treat (DTT) pathogens in 16 infections (7.5%) [all without rifampicin-resistant strains] and methicillin-resistant staphylococci (MRS) in 24 infections (11.3%).

### 4.3. Treatment Protocol

The diagnosis of a periprosthetic infection was determined according to the MSIS and ICM criteria with joint aspiration and/or joint biopsy performed. A two-stage septic exchange procedure was performed in cases of periprosthetic joint infection. In the stage-one surgery, explantation of the infected endoprosthesis, radical debridement and the implantation of spacer components were performed. Furthermore, samples for microbiological and histological examination were taken intraoperatively. The Krenn and Morawietz type classification [24] was used for the histological examination of the synovial membrane and the periprosthetic tissue. At the hip, the spacer consisted of a hip stem and a cemented cup coated with individualized multiantibiotic-loaded bone cement, as previously described [4,30,31]. The bone cement was tailored according to the resistance profile of the pathogen isolated preoperatively. At the knee, there were 23 mobile and 60 static spacers (in knees with ligamentous instability).

The postoperative antibiotic therapy was the recommendation of a microbiological consultant and in accordance with the individual antibiogram. An initial parenteral antibiotic therapy for two weeks was followed by an oral antibiotic therapy for four weeks (Table 2 and Table 3). The systemic antibiotic therapy with rifampicin (600 mg/day) started on the second postoperative day (oral mostly on the third postoperative day) because it could be expected that at that time the companion antibiotic medication to which the bacterial isolate was susceptible (started immediately during revision surgery) had reached the therapeutic concentration. By that, a monotherapy with rifampicin, with the high risk of development of resistance, was prevented. The stage-two surgery took place after an interim period of 6 weeks and included explantation of the spacer components, repeated radical debridement and reimplantation of an endoprosthesis. The same systemic antibiotic therapy was given after reimplantation, with 2 weeks i.v. and 4 weeks orally. Rifampicin was started again on the second day postoperatively.

### 4.4. Tests at Second Stage Revision Surgery

During both revision surgeries (explantation and reimplantation), samples for microbiological examination were extracted from five distinct areas close to the prosthesis (periprosthetic tissue and synovium). Additionally, five samples from the synovium and periprosthetic connective tissue membrane associated with the loosened prosthesis were procured for histological evaluation. To prevent the distortion of microbiological results, the administration of perioperative antibiotics took place after the sample collection. The samples for microbiological analysis were placed in sterile sample tubes and were sent to the microbiological institute within a one-hour timeframe. The same procedure was used for synovial fluid samples obtained pre-operatively.

The microbiological samples were inoculated onto blood agar and into nutrient broth specific to anaerobic pathogens. The incubation period was 14 days [32].

The pathogens were categorized into three groups according to Faschingbauer et al. [33]. Group 1 consists of difficult-to-treat (DTT) pathogens, quinolone-resistant Gram-negative bacteria, enterococcus and candida. Methicillin-resistant staphylococci (MRS), including methicillin-resistant staphylococcus aureus (MRSA) and methicillin-resistant staphylococcus epidermidis (MRSE) are classed as group 2. Group 3 consists of easy-to-treat (ETT) pathogens, encompassing any remaining pathogen and culture-negative periprosthetic joint infections.

In histological analysis, the classification system of Morawietz and Krenn et al. was used to categorize the periprosthetic tissue [24,25,26]. The periprosthetic tissue was differentiated into four types: the particle type (I), the infection type (II), the combined type (III) and the indeterminate type (IV). The histopathologist also assessed the number of polymorphonuclear leukocytes per high-power microscope field.

The microbiological assessment of the tissue samples and the analysis of the aspiration fluids were carried out according to the ICM criteria [22,23]: this means that, if the cumulative diagnostic score was equal to or greater than 6, a periprosthetic joint infection (PJI) was diagnosed.

Regular follow-up examinations took place for at least two years. Patients were classified as free of reinfection according to Diaz-Ledezma et al. [34]: microbiological and clinical absence of infection, no subsequent further surgical intervention for PJI and no mortality related to PJI. In cases suspicious of PJI, MSIS criteria 2014 and ICM criteria 2018 were applied [23].

### 4.5. Statistical Analyses

IBM SPSS Statistics for Windows (version 27, IBM Crop., Armonk, NY, USA) was used for statistical analysis. To calculate group differences, the univariate analysis of variance was used. Unless otherwise stated, data are presented as mean ± standard deviation or number (percentage). The significance level was set at *p* < 0.05.

The research was performed following the guidelines of the Declaration of Helsinki. The study protocol was approved by the Ethics Committee of Landesaerztekammer Baden-Württemberg (committee’s reference number F-2023-115).

## 5. Conclusions

In this study, only a small percentage of 0.9% after the first stage and 1.4% at follow-up after the second stage of all 212 patients with accompanying long-term rifampicin combination therapy developed a rifampicin resistance. Therefore, in our opinion, rifampicin administration can be started on the second day postoperatively (when the concentration of the accompanying antibiotics is sufficiently high) without increasing the risk of rifampicin resistance. The benefit would be antibiotic therapy, beginning when the biofilm of any remaining bacteria is relatively young and can be penetrated by rifampicin.

## Figures and Tables

**Figure 1 antibiotics-14-00610-f001:**
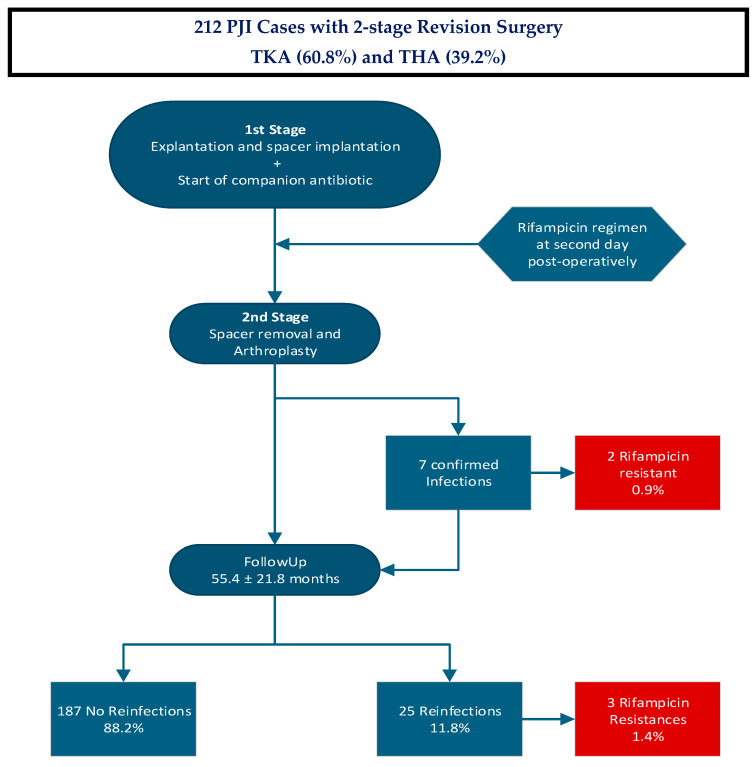
The schematic study course visualizes the different stages of the described study: A total of 212 patients with diagnosed PJI received first-stage surgery explantation, debridement and spacer implantation. The specific companion medication started immediately and was followed by the rifampicin regimen from the second postoperative day. All 212 patients received spacer explantation and arthroplasty. At that time, seven infections were confirmed, of which two showed rifampicin-resistant strains. At follow-up after reimplantation, a total of 187 patients showed no reinfections, whereas 25 patients had reinfections, of which, three showed rifampicin-resistant strains.

**Table 1 antibiotics-14-00610-t001:** Demographic data of study population.

	Sum	Age	BMI	Diabetes	Rheum. Disorders	ASA Risk Classification			
	N (%)	M (SD)	M (SD)	N (%)	N (%)	ASA1	ASA2	ASA3	ASA4
TKA	83 (39.2%)	70.29 (11.41)	32.22 (7.75)	20 (24.1%)	0 (0.0%)	4 (4.8%)	34 (41.0%)	42 (50.6%)	3 (3.6%)
THA	129 (60.8%)	68.50 (11.32)	29.28 (6.12)	24 (18.6%)	10 (7.8%)	1 (0.8%)	60 (46.5%)	65 (50.4%)	3 (2.3%)
Total	212 (100%)	69.20 (11.36)	30.43 (6.94)	44 (20.8%)	10 (4.7%)	5 (2.4%)	94 (44.3%)	107 (50.5%)	6 (2.8%)

**Table 2 antibiotics-14-00610-t002:** Intravenously administered antibiotics or antimycotics and number.

Antibiotic One	Antibiotic Two	Antibiotic Three	Number
Vancomycin	Rifampicin		58
Flucloxacillin	Rifampicin		46
Ampicillin/Sulbactam	Rifampicin		43
Cefuroxim	Rifampicin		12
Levofloxacin	Rifampicin		11
Imipenem	Rifampicin		10
Penicillin G	Rifampicin		9
Daptomycin	Rifampicin		7
Amoxicillin/Clavulanacid	Rifampicin		4
Ampicillin	Rifampicin		3
Vancomycin	Meropenem	Rifampicin	3
Imipenem/cilastatin	Rifampicin		2
Vancomycin	Imipenem	Rifampicin	2
Flucloxacillin	Levofloxacin	Rifampicin	1
Flucloxacillin	Meropenem	Rifampicin	1

**Table 3 antibiotics-14-00610-t003:** Administered oral antibiotics or antimycotics and number.

Antibiotic One	Antibiotic Two	Antibiotic Three	Number
Levofloxacin	Rifampicin		141
Amoxicillin/Clavulanacid	Rifampicin		31
Cotrimoxazol	Rifampicin		13
Linezolid	Rifampicin		9
Clindamycin	Rifampicin		7
Ampicillin/Sulbactam	Rifampicin		4
Moxifloxacin	Rifampicin		3
Clarythromycin	Levofloxacin	Rifampicin	2
Amoxicillin/Clavulanacid	Levofloxacin	Rifampicin	1
Cotrimoxazol	Levofloxacin	Rifampicin	1

## Data Availability

We do not wish to share our data because some of the patient’s data relate to individual privacy and, according to the policy of our hospital, these data may not be shared with others without permission.

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
