# Peer review of "Early Administration of Rifampicin Does Not Induce Increased Resistance in Septic Two-Stage Revision Knee and Hip Arthroplasty"

_antibiotics, 2025, doi:10.3390/antibiotics14060610_

Round 1
Reviewer 1 Report
Comments and Suggestions for Authors
The authors carried out a retrospective study assessing the prevalence of rifampicin resistance and its characteristics. Following comments may be useful to authors:
- Rifampicin was administered in combination with other antimicrobial drugs. What are the combinations observed following oral and intravenous administrations?
- Could sub-group analyses be carried out between different combinations?
- Are there data regarding the patients compliance following oral therapy? How does this influence the development of resistance in your patient cohort?
- Were there any comorbid immunosuppressive disordes such as poorly controlled diabetes or other concomitant medicines that could potentially induce the development of resistance in your patient cohort?
- How was the sample size estimated in your study?
- Please adhere to EQUATOR reporting guidelines for retrospective studies.
- Provide a checklist filling the page numbers related to each of the items in the reporting guidelines.
- In the discussion, provide the strengths of your study with future directions to the practicing clinicians and researchers.
Author Response
Reviewer 1:
Thank you very much for reviewing our paper and the helpful comments. Are the comments are addressed in new version. The answers to the comments are here in red and the changes in the manuscript also in red.
The authors carried out a retrospective study assessing the prevalence of rifampicin resistance and its characteristics. Following comments may be useful to authors:
- Rifampicin was administered in combination with other antimicrobial drugs. What are the combinations observed following oral and intravenous administrations? Answer: This is added in Table 2 and 3
- Could sub-group analyses be carried out between different combinations? Answer: Because of the small numbers of restistance a aub-group analyses could not be performed. This is mentioned as a limitation in the discussion section.
- Are there data regarding the patients compliance following oral therapy? How does this influence the development of resistance in your patient cohort? Answer: There are no real data regarding patient compliance. We only think that the patients took the medication.
- Were there any comorbid immunosuppressive disordes such as poorly controlled diabetes or other concomitant medicines that could potentially induce the development of resistance in your patient cohort? Answer: The comorbidities are listet in Table 1 and the Charlson Comobidity Index is mentioned below the table 1.
- How was the sample size estimated in your study? Answer: A power analysis was made, assuming a rate of resistance around 10 %.
- Please adhere to EQUATOR reporting guidelines for retrospective studies. Answer: This is done
- Provide a checklist filling the page numbers related to each of the items in the reporting guidelines. Answer: the checklist is added at the end. The organization of the different sections is according to the style of the journal.
- In the discussion, provide the strengths of your study with future directions to the practicing clinicians and researchers. Answer: is done at the end of the discussion section
- STROBE Statement—Checklist of items that should be included in reports of cohort studies
|
|
Item No |
Recommendation |
Lines No |
|
Title and abstract |
1 |
(a) Indicate the study’s design with a commonly used term in the title or the abstract |
1-4 |
|
(b) Provide in the abstract an informative and balanced summary of what was done and what was found |
13-31 |
||
|
Introduction |
|||
|
Background/rationale |
2 |
Explain the scientific background and rationale for the investigation being reported |
34-62 |
|
Objectives |
3 |
State specific objectives, including any prespecified hypotheses |
63-79 |
|
Methods |
|||
|
Study design |
4 |
Present key elements of study design early in the paper |
224-227 |
|
Setting |
5 |
Describe the setting, locations, and relevant dates, including periods of recruitment, exposure, follow-up, and data collection |
228-235 |
|
Participants |
6 |
(a) Give the eligibility criteria, and the sources and methods of selection of participants. Describe methods of follow-up |
233-269 |
|
(b) For matched studies, give matching criteria and number of exposed and unexposed |
|
||
|
Variables |
7 |
Clearly define all outcomes, exposures, predictors, potential confounders, and effect modifiers. Give diagnostic criteria, if applicable |
311-341 |
|
Data sources/measurement |
8* |
For each variable of interest, give sources of data and details of methods of assessment (measurement). Describe comparability of assessment methods if there is more than one group |
247-277, 279-341 |
|
Bias |
9 |
Describe any efforts to address potential sources of bias |
228-244 |
|
Study size |
10 |
Explain how the study size was arrived at |
227-232 |
|
Quantitativevariables |
11 |
Explain how quantitative variables were handled in the analyses. If applicable, describe which groupings were chosen and why |
247-255 |
|
Statistical methods |
12 |
(a) Describe all statistical methods, including those used to control for confounding |
345-353 |
|
(b) Describe any methods used to examine subgroups and interactions |
|
||
|
(c) Explain how missing data were addressed |
|
||
|
(d) If applicable, explain how loss to follow-up was addressed |
|
||
|
(e) Describe any sensitivity analyses |
|
||
|
Results |
|
||
|
Participants |
13* |
(a) Report numbers of individuals at each stage of study—eg numbers potentially eligible, examined for eligibility, confirmed eligible, included in the study, completing follow-up, and analysed |
82-91 |
|
(b) Give reasons for non-participation at each stage |
|
||
|
(c) Consider use of a flow diagram |
114 |
||
|
Descriptive data |
14* |
(a) Give characteristics of study participants (eg demographic, clinical, social) and information on exposures and potential confounders |
93-108 |
|
(b) Indicate number of participants with missing data for each variable of interest |
|
||
|
(c) Summarise follow-up time (eg, average and total amount) |
|
||
|
Outcome data |
15* |
Report numbers of outcome events or summary measures over time |
87-99, 106-108 |
|
Main results |
16 |
(a) Give unadjusted estimates and, if applicable, confounder-adjusted estimates and their precision (eg, 95% confidence interval). Make clear which confounders were adjusted for and why they were included |
93-100, 106-113 |
|
(b) Report category boundaries when continuous variables were categorized |
|
||
|
(c) If relevant, consider translating estimates of relative risk into absolute risk for a meaningful time period |
|
||
|
Other analyses |
17 |
Report other analyses done—eg analyses of subgroups and interactions, and sensitivity analyses |
101-105, 122-127 |
|
Discussion |
|||
|
Key results |
18 |
Summarise key results with reference to study objectives |
130-133 |
|
Limitations |
19 |
Discuss limitations of the study, taking into account sources of potential bias or imprecision. Discuss both direction and magnitude of any potential bias |
210-222 |
|
Interpretation |
20 |
Give a cautious overall interpretation of results considering objectives, limitations, multiplicity of analyses, results from similar studies, and other relevant evidence |
134-208 |
|
Generalisability |
21 |
Discuss the generalisability (external validity) of the study results |
355-361 |
|
Other information |
|||
|
Funding |
22 |
Give the source of funding and the role of the funders for the present study and, if applicable, for the original study on which the present article is based |
369 |
Reviewer 2 Report
Comments and Suggestions for Authors
The manuscript can benefit from a clearer structural distinction between primary outcomes (rifampicin resistance) and secondary (reinfection rates) as the current results section tends to blend these which reduce clarity for readers. The discussion does not sufficiently address variables (comorbidities, previous revisions, differences in pathogen profiles) which could influence outcomes. The limitations of a retrospective design regarding data completeness and lack of a control group are acknowledged only briefly and need a more analysis to add to the strength of the conclusions. Several phrases in the manuscript are awkward or incorrect grammar ("continuously infection parameters") and a thorough language revision is necessary. Some references show minor formatting inconsistencies, and a detailed citation check is recommended. While the clinical relevance is ok the conclusion section does not translate the findings into practical recommendations or changes in clinical guidelines/ decisions. The title is relevant.
Comments on the Quality of English LanguageSeveral phrases in the manuscript are awkward or incorrect grammar ("continuously infection parameters") and a thorough language revision is necessary.
Author Response
Reviewer 2:
Thank you very much for reviewing our paper and the helpful comments. Are the comments are addressed in new version. The answers to the comments are here in red and the changes in the manuscript also in red.
The manuscript can benefit from a clearer structural distinction between primary outcomes (rifampicin resistance) and secondary (reinfection rates) as the current results section tends to blend these which reduce clarity for readers. Answer: The result section is rewritten in this way
The discussion does not sufficiently address variables (comorbidities, previous revisions, differences in pathogen profiles) which could influence outcomes. Answer: This is mentioned as a limitation in the last paragraph of the discussion section.
The limitations of a retrospective design regarding data completeness and lack of a control group are acknowledged only briefly and need a more analysis to add to the strength of the conclusions. Answer: This has been discussed in more detail in the limitations in the last paragraph of the discussion section.
Several phrases in the manuscript are awkward or incorrect grammar ("continuously infection parameters") and a thorough language revision is necessary. Answer: The paper has been reviewed by a nativ speaking scientist.
Some references show minor formatting inconsistencies, and a detailed citation check is recommended. Answer: The references have been corrected
While the clinical relevance is ok the conclusion section does not translate the findings into practical recommendations or changes in clinical guidelines/ decisions. Answer: This has been described in more detail in the conclusion section.
The title is relevant. Answer: No comment necessary
Reviewer 3 Report
Comments and Suggestions for Authors
Following are my comments and suggestions that the author should address to improve the manuscript:
-
Line 16 – The term "combination therapy" is unclear and potentially confusing. Please clarify or provide additional context.
-
Figure 1 – The figure legend titled "Schematic study course" needs further elaboration. Please describe the elements and flow of the schematic in more detail to enhance reader understanding.
-
It is unclear whether the study was conducted on a state-wise or country-wise basis. Please specify the geographical scope of the study.
-
The manuscript focuses solely on rifampicin. Please justify this choice, considering that there are several other antibiotics and antifungals that can also be administered to patients.
-
The dosage of rifampicin should be discussed in the manuscript, as it is a critical factor influencing the development of antibiotic resistance mechanisms in pathogens.
-
A comparative analysis with other antibiotics would strengthen the study. Please consider including a comparison to support the relevance and effectiveness of rifampicin in this context.
Author Response
Reviewer 3:
Thank you very much for reviewing our paper and the helpful comments. Are the comments are addressed in new version. The answers to the comments are here in red and the changes in the manuscript also in red.
Following are my comments and suggestions that the author should address to improve the manuscript:
- Line 16 – The term "combination therapy" is unclear and potentially confusing. Please clarify or provide additional context. Answer: This has been rewritten
- Figure 1 – The figure legend titled "Schematic study course" needs further elaboration. Please describe the elements and flow of the schematic in more detail to enhance reader understanding. Answer: The figure legend has been rewritten
- It is unclear whether the study was conducted on a state-wise or country-wise Please specify the geographical scope of the study. Answer: This has been described at the beginning of the material and method section in more detail.
- The manuscript focuses solely on rifampicin. Please justify this choice, considering that there are several other antibiotics and antifungals that can also be administered to patients. Answer: This was already mentioned in lines 50 and 56 and has been mentioned in line 250 as well as in the limitations of the discussion section in more detail.
- The dosage of rifampicin should be discussed in the manuscript, as it is a critical factor influencing the development of antibiotic resistance mechanisms in pathogens. This is mentioned in more detail in the discussion section.
- A comparative analysis with other antibiotics would strengthen the study. Please consider including a comparison to support the relevance and effectiveness of rifampicin in this context. Answer: This is mentioned in the limitations of the discussion section
Round 2
Reviewer 1 Report
Comments and Suggestions for Authors
Thanks for the revision.
Reviewer 3 Report
Comments and Suggestions for Authors
Accept